# Registers in Small Vision Transformers: A Reproducibility Study of Vision Transformers Need Registers

**Linus Bach**[†]                                              *linus.bach@student.uva.nl*
*University of Amsterdam*

**Emma Bakker**[†]                                             *emma.bakker3@student.uva.nl*
*University of Amsterdam*

**Rénan van Dijk**[†]                                          *renan.van.dijk@student.uva.nl*
*University of Amsterdam*

**Jip de Vries**[†]                                            *jip.devries@student.uva.com*
*University of Amsterdam*

**Konrad Szewczyk**                                            *konszewczyk@outlook.com*
*University of Amsterdam*

[†] *These authors contributed equally to this work.*

**Reviewed on OpenReview:** *https://openreview.net/forum?id=5JflRlCt3Q*

## Abstract

Recent work has shown that Vision Transformers (ViTs) can produce "high-norm" artifact tokens in attention maps. These artifacts disproportionately accumulate global information, can degrade performance, and reduce interpretability in these models. Darcet et al. (2024) proposed registers—auxiliary learnable tokens—to mitigate these artifacts. In this reproducibility study, we verify whether these improvements extend to smaller ViTs. Specifically, we examine whether high-norm tokens appear in a DeiT-III Small model, whether registers reduce these artifacts, and how registers influence local and global feature representation. Our results confirm that smaller ViTs also exhibit high-norm tokens and registers partially alleviate them, improving interpretability. Although the overall performance gains are modest, these findings reinforce the utility of registers in enhancing ViTs while highlighting open questions about their varying effectiveness across different inputs and tasks. Our code is available at `https://github.com/SnorrenanxD/regs-small-vits`.

## 1 Introduction

Vision transformers (ViTs) have emerged as a powerful alternative to convolutional neural networks (CNNs) in computer vision, offering a flexible mechanism for modeling spatial relationships in images. Originally introduced by Dosovitskiy et al. (2021), ViTs use self-attention mechanisms to capture both local and global interactions within images, enabling strong performance across tasks such as image classification, segmentation, and detection.

Building on the foundational ViT architecture, advancements like DeiT (Touvron et al., 2021), DeiT-III (Touvron et al., 2022), and the DINOv2 model (Oquab et al., 2023) have significantly enhanced the efficiency and effectiveness of ViTs. DeiT and DeiT-III introduced refined training methodologies including new augmentation techniques and optimized regularization strategies, making ViTs more data-efficient and practical for datasets like ImageNet-1k and ImageNet-21k (Deng et al., 2009). Concurrently, DINOv2 rein-

forced the role of self-supervised learning in ViTs by generating robust, general-purpose visual features from large-scale curated datasets, enabling superior performance in both image-level and pixel-level tasks.

Despite these advancements, ViTs, including enhanced versions like DeiT and DINOv2, face significant challenges. Darcet et al. (2024) identified the presence of artifacts in the self-attention maps of transformer blocks, which encapsulate global information and inadvertently lead to the loss of crucial local details. The authors argue that traditional ViTs face two significant limitations: (1) a heavy reliance on transformer layers to encode all image-level information, which can lead to inefficiencies, and (2) the lack of explicit mechanisms for maintaining state information across layers, potentially affecting long-term coherence in representations.

To address these limitations, Darcet et al. (2024) introduced the concept of registers—special tokens inspired by computer architecture that accumulate and carry information between layers—as an enhancement to the ViT framework. By functioning as memory units within the transformer sequence, registers provide a structured mechanism for managing both local and global information more effectively. This approach reduces reliance on transformer layers to encode all image-level information and offers explicit state maintenance across layers. Integrating registers has been shown to improve interpretability and feature smoothness by mitigating inconsistencies in feature maps and balancing local and global contexts more effectively.

The introduction of additional tokens for the purpose of retaining abstract information to transformer architectures has been proven effective in the field of NLP. Most relevant for our investigation is the work of Burtsev et al. (2020), who showed that concatenating *memory tokens* to a series of token embeddings improves translation quality in transformer-based machine translation models. The authors observe a similar issue to the artifacts in ViTs when mixing local and global information within token embeddings, inevitably leading to inefficiencies. While the authors extend their analysis to separate memory controllers, their results indicate that simply enabling the model to isolate global sequence representations improves performance and interpretability.

While prior work on registers has focused on large-scale Vision Transformers (ViT-L and above), recent studies show that high-norm artifact tokens are not exclusive to these architectures. Darcet et al. (2024) found these artifacts to be especially prominent in large models (see their Figure 4c), but follow-up work has observed similar behavior in smaller ViTs, albeit typically less frequently or less severely (Nakamura et al., 2024; Shang et al., 2024). This suggests that such artifacts represent a broader phenomenon in ViTs. Since smaller models are widely used in real-time and resource-constrained settings, mitigating artifacts in this regime is both theoretically and practically relevant. If registers improve local feature fidelity and attention consistency in small models, they offer a lightweight enhancement worth exploring. Conversely, if artifact severity scales with model size, this helps define when registers are most beneficial.

In this reproducibility study, we examine the main claims from Darcet et al. (2024) and extend their experiments to smaller-scale models. Specifically, we explore whether (1) high-norm artifact tokens are also present in a smaller ViT, (2) registers can mitigate them effectively and (3) the improvements in local-global encoding reported for larger models also generalize to smaller architectures. By replicating and expanding these experiments, we aim to clarify the broader relevance of registers in stabilizing and refining Vision Transformer representations.

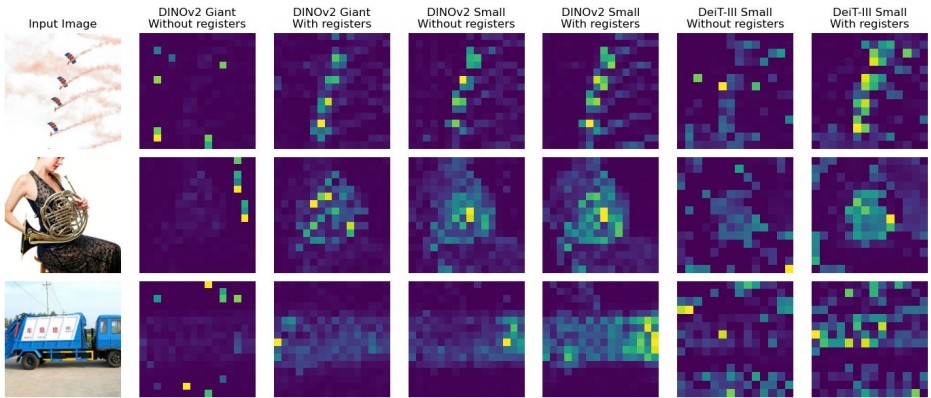

Figure 1: A comparison of attention maps from the DINOv2 giant model with and without register tokens, the DINOv2 small model with and without register tokens and our DeiT-III small models with and without register tokens. Register tokens enable interpretable attention maps for most images of the DINOv2 model, while they do not necessarily improve the attention maps of the DeiT-III model.

## 1.1 Key Concepts and Terminology

We briefly define key terms used throughout this study to ensure clarity.

**High-norm tokens** (or *artifact tokens*) are token embeddings with unusually large norms, often encoding global information at the expense of local detail. In attention heatmaps, they appear as *bright* (yellow or green) patches, as seen in the first column of Figure 1.

**Low-norm tokens** have more typical norms and retain more local features. They correspond to *darker* (blue or purple) regions in the maps.

**Registers** are additional learnable tokens inserted into the input sequence. They help carry global context across layers and reduce the burden on patch tokens. This mitigates high-norm artifacts and results in more evenly distributed attention, as shown in the rightmost columns of Figure 1.

## 2 Scope of Reproducibility

In their study, Darcet et al. (2024) use ground-up-trained DeiT-III base, OpenCLIP, and DINOv2 large models. We replicate these original experiments on larger pretrained ViT models and extend the investigation to a smaller DeiT-III Small model, trained from scratch on ImageNet-1k. Our focus is on determining whether high-norm tokens persist across different architectures and scales and whether registers consistently alleviate these anomalies while improving learned representations. By exploring smaller ViTs, we assess whether the benefits observed in previous work hold in more constrained architectures.

### 2.1 Model Architecture and Registers Implementation

Artifact tokens, visible as strong peaks in self-attention maps (Figure 1), indicate where the model allocates disproportionate attention, potentially destabilizing its representation. Additionally, these outliers can be identified as having a significantly raised token norm. Throughout this work, we will use the terms high-norm tokens and artifacts synonymously.

Darcet et al. (2024) addressed this by introducing *registers*—additional learnable tokens that act as intermediate memory, reducing the transformer's reliance on attention layers alone for global context. In that work, registers were added to a DINOv2-based ViT (see Figure 2 of Darcet et al., 2024), where they join the input sequence alongside the [CLS] and patch tokens. This approach mitigated high-norm artifacts and improved representations, particularly in dense tasks where patch tokens are retained.

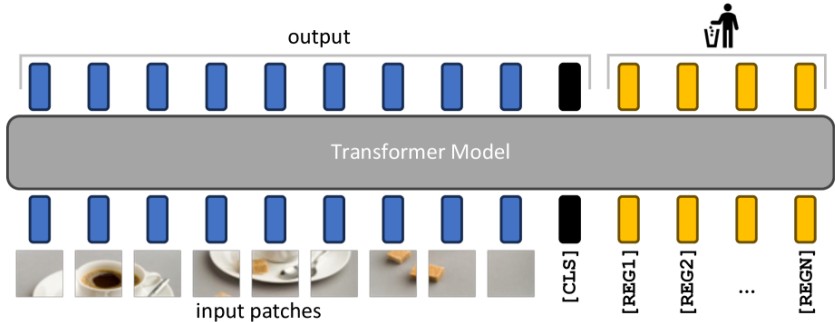

Figure 2: Illustration of the proposed remediation and resulting model. We add *N* additional learnable input tokens (depicted in yellow), that the model can use as registers. At the output of the model, only the patch tokens and [CLS] tokens are used, both during training and inference. *Adapted from Darcet et al. (2024).*

In our study, we adopt the same core idea of the original paper by adding registers to DeiT-III models. Specifically, we investigate whether a DeiT-III Small architecture benefits similarly from these additional tokens. We focus on confirming the presence of high-norm tokens in smaller models and evaluating whether registers effectively resolve or reduce these artifacts.

## 2.2   Main Claims

The original paper by Darcet et al. (2024) highlights three key points that we seek to reproduce and validate in this study:

1. **High-Norm Tokens (Artifacts) in Larger Models:** High-norm "artifact" tokens consistently appear in large-scale ViTs.

2. **Register-Based Mitigation:** Introducing registers—additional learnable tokens acting as intermediate memory—reduces or eliminates these high-norm artifacts, leading to more stable and interpretable attention distributions.

3. **Enhanced Local-Global Encoding:** High-norm tokens reflect an imbalance in how local and global information is captured. By segregating global information into registers, ViTs retain a more effective balance of local and global features, resulting in more coherent and expressive intermediate representations.

Together, these claims propose that registers offer a robust architectural enhancement for ViTs, improving both interpretability and representation quality and potentially generalizing across different model scales and training paradigms.

## 2.3   Broader Context

The incorporation of registers in ViTs makes these models more interpretable and efficient. By reducing inconsistencies in feature and attention maps, the registers clarify the model's internal processes, enhancing trust and understanding. They also streamline information flow, potentially lowering computational overhead and improving scalability, which is a crucial advantage in resource-constrained settings. By incorporating smaller-scale DeiT-III models into this study, we broaden the scope of Darcet et al. (2024) to examine whether their proposed register mechanism also benefits ViTs with fewer parameters. If confirmed, these findings suggest that registers can offer a robust architectural enhancement to a wider array of ViT variants, potentially leading to more interpretable and efficient models across different scales.

# 3 Experiments

To address the first two claims outlined above, we conducted experiments on pretrained DINOv2 models as well as DeiT-III models trained from scratch, with and without registers. Our primary analyses include examining attention maps, token norms and final performance metrics on ImageNet-1k.

To evaluate claim 3—concerning the models' handling of global information—we first analyze the spatial token embeddings for global content and then apply the LOST method (Siméoni et al., 2021) to assess object-level understanding. LOST (Localizing Objects with Self-supervised Transformers) is a self-supervised object discovery method that uses patch token similarity to identify salient regions in an image without any labeled data. It relies on local relationships between patch representations and provides insight into the emergence of object-level semantics in transformer models. We apply LOST to our DeiT-III model with register tokens to assess whether the inclusion of registers indirectly improves the model's ability to localize objects by influencing the quality of patch-level representations.

## 3.1 Model descriptions

In this study, we evaluated the impact of incorporating registers in ViTs using both pretrained DINOv2 models as well as DeiT-III models that were trained from scratch.

**Pretrained DINOv2 Models**: For the reproducibility part of our study, we analyzed the pretrained DINOv2 Large model (300M parameters), obtained from the official repository[1]. DINOv2 employs a vanilla ViT backbone and a dual-network self-distillation setup, in which one network generates target embeddings that the other learns. We evaluated the large model with and without registers to replicate key results from Darcet et al. (2024). We also performed diagnostic checks on the smaller DINOv2 variant (21M parameters) to assess whether high-norm artifact tokens appear at a reduced scale.

**DeiT-III Models**: We also trained DeiT-III Small models (22M parameters) from scratch to directly measure the impact of register tokens on performance and representation. All models were trained for 800 epochs on ImageNet-1k using $16 \times 16$ patches and standardized with ImageNet mean-std normalization, following the official DeiT-III repository by Touvron et al. (see Appendix A for details). To evaluate the effect of register count, we trained models with 0, 2, 4, 8, and 16 register tokens, implemented as additional CLS-style tokens following the DINOv2 setup. Unless otherwise stated, we use the 4-register configuration for qualitative analyses, to remain consistent with the setting used by Darcet et al. (2024).

## 3.2 Datasets

This study primarily uses ImageNet-1k (Deng et al., 2009) for training and evaluation [2]. ImageNet-1k contains 1.28 million training images and 50,000 validation images across 1,000 classes. We use the official dataset split, which resizes images to $224 \times 224$ pixels and applies ImageNet mean-std normalization.

For the LOST experiment, we use the VOC07 dataset (Everingham et al., 2010). Additionally, we employ Imagenette (Torii, 2020) and Caltech101 (Li et al., 2022) for further evaluation of token representations in specific experiments.

## 3.3 Experimental setup and code

We integrated the register tokens into the DeiT-III codebase by inserting additional learnable tokens alongside the [CLS] and patch tokens, as shown in Figure 3. The figure illustrates the configuration with four registers, following Darcet et al. (2024), but in our experiments we evaluated models with 2, 4, 8, and 16 register tokens. All registers were randomly initialized from a normal distribution with low variance, following the same approach as for the [CLS] token. Unlike DINOv2-based architectures, our DeiT-III setup discards all patch tokens at the output stage, relying solely on the [CLS] token for classification.

---

[1]https://github.com/facebookresearch/dinov2/tree/main
[2]https://image-net.org/download-images

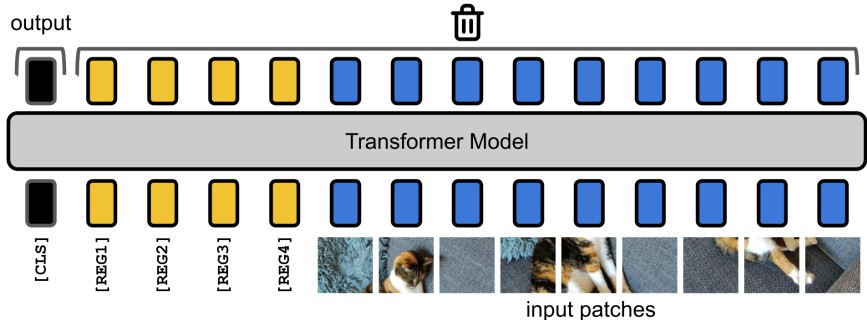

Figure 3: Modified ViT architecture with registers. The 4 additional learnable input tokens (highlighted in yellow) act as registers, while only the [CLS] token is outputted during training and inference.

## 4 Results

This section examines our results in relation to the three main claims from Section 2: (1) the presence of high-norm tokens, (2) the effectiveness of registers in reducing artifacts, and (3) their impact on local-global feature balance. We structure our findings as follows: Section 4.1 evaluates *classification performance*, Section 4.2 analyzes *local features* by measuring high-norm tokens in DINOv2 and DeiT-III, and Section 4.3 investigates *global features*, using LOST for object-level understanding and assessing spatial patch token representations.

### 4.1 Classification Performance

We begin by evaluating our models on the ImageNet-1k (Deng et al., 2009) validation set. The baseline DeiT-III Small model (without registers) achieves a top-1 accuracy of 81.05%. When two register tokens are added, accuracy slightly decreases to 80.96%, and with four registers it further drops to 80.81%. However, when using eight and sixteen registers, accuracy recovers to 81.04% and 81.07%, respectively. These differences are small, indicating that register count has a minimal effect on classification accuracy. For the rest of the analyses in this paper, we use the four-register configuration to enable direct comparison with the DINOv2-based setup in Darcet et al. (2024).

### 4.2 Local Features

#### 4.2.1 Local information of high-norm tokens

To test whether high-norm tokens primarily carry detailed local information or global context, we train a small Multi-Layer Perceptron (MLP) to reconstruct the original $16 \times 16 \times 3$ image patch from a single token embedding. The MLP has an input size of 384 (matching our DeiT-III hidden dimension), a single hidden layer of size 1536, and an output layer of 768 (for the flattened $16 \times 16 \times 3$ patch). We train this MLP for five epochs on high- and low-norm tokens drawn from our DeiT-III model with no register tokens.

We find that high-norm tokens exhibit a substantially larger MSE of 176.54 compared to 120.02 for low-norm tokens, representing a 147.1% increase in error. This indicates that high-norm tokens lose a significant amount of local detail, supporting the original claim by Darcet et al. (2024) that high-norm "artifact" tokens predominantly capture global information and thus encode fewer spatial details.

#### 4.2.2 Artifacts in DINOv2

Unless otherwise stated, all token norm and cosine similarity analyses in this section are computed over the full Imagenette validation set (3925 images). Using this data, we observe how token norms evolve in the pretrained DINOv2 giant model. In Figure 4a, the distribution of token norms for the DINOv2 giant model without registers displays a heavy tail toward large values. These "outlier" tokens, which have notably higher norms than the rest, coincide with the "artifacts" identified by Darcet et al. (2024), indicating an

accumulation of excessive global information in just a few tokens. Such imbalanced allocation of global context may adversely affect the encoding of local features.

When registers are introduced (Figure 4b), the long tail of the distribution shrinks significantly, signaling that fewer tokens are carrying disproportionately large norms. This aligns with the hypothesis that registers may help absorb or redistribute global context that would otherwise concentrate in certain patch tokens, although our results do not directly establish causality. Consequently, no single token disproportionately dominates the self-attention mechanism and the overall norm distribution becomes more balanced.

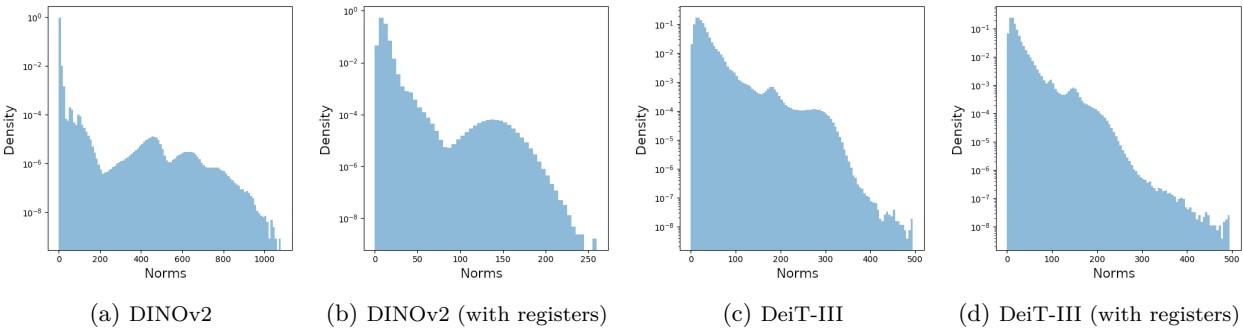

(a) DINOv2    (b) DINOv2 (with registers)    (c) DeiT-III    (d) DeiT-III (with registers)

Figure 4: Histograms of token norm distributions for DINOv2 giant (subfigures a–b) and DeiT-III Small (subfigures c–d), comparing models without registers (left panels) and with registers (right panels).

Looking more closely at how these norms progress across the network, Figure 5 shows layer-wise norm distributions for DINOv2 giant. Without registers (left), the deeper layers produce outlier tokens that far exceed the typical norm range, indicating an over-accumulation of global context. With registers (right), the distribution becomes smoother and high-norm tokens effectively vanish, reflecting a more balanced distribution of information.

Qualitative attention maps (Figure 1) corroborate these findings. Without registers, the model's attention is sharply focused on a few regions, reinforcing the notion that artifact tokens become dominant. In contrast, the inclusion of registers yields more uniformly distributed and interpretable attention. Collectively, these observations validate that registers effectively mitigate high-norm artifacts in the DINOv2 giant architecture, ensuring a healthier balance between local and global feature representation.

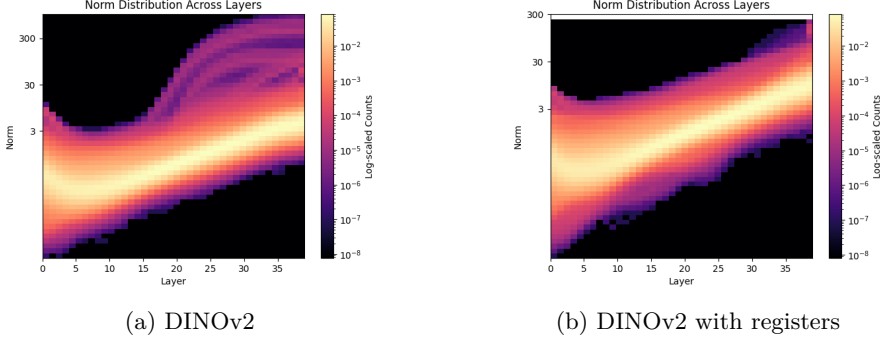

(a) DINOv2           (b) DINOv2 with registers

Figure 5: Distribution of norm values across layers for models. Registers eliminate high-norm outliers in deeper layers.

Finally, we replicate the cosine similarity analysis for DINOv2 giant. In Figure 6a, we measure the cosine similarity of the initial patch encodings of neighboring tokens. Here, we set the threshold for high-norm (artifact) tokens at the 98th percentile norm value, following Darcet et al. who found approximately 2.37% of tokens to be outliers. Consistent with Darcet et al. (2024), we observe that these high-norm tokens tend to be more internally consistent with some of their neighbors (i.e., close to a cosine similarity of 1.0), suggesting

a strong clustering of attention in those areas. The introduction of registers helps redistribute this global context more evenly, diminishing the outlier effect and producing more stable representations.

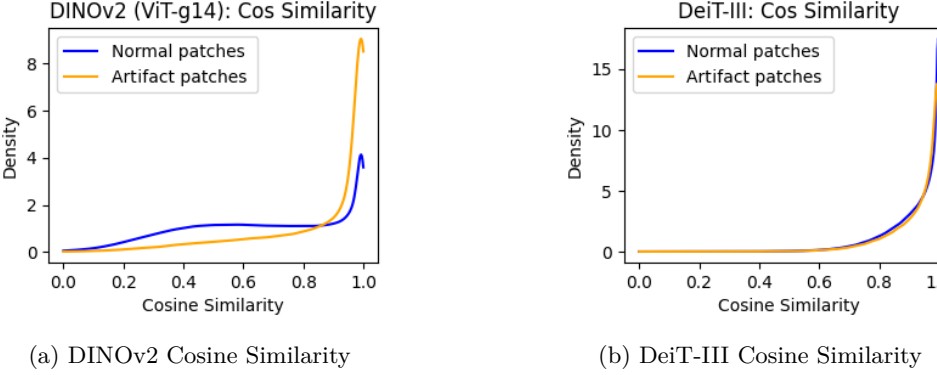

(a) DINOv2 Cosine Similarity

(b) DeiT-III Cosine Similarity

Figure 6: Cosine similarity analysis for DINOv2 and DeiT-III models.

Figure 1 shows that the small DINOv2 models do not demonstrate artifacts in the classification token's attention maps. This supports the original paper's claim that these artifacts only appear in larger models. We have noted that this does not hold for the DeiT-III model, where artifacts do arise in the small version of the model.

### 4.2.3 Artifacts in DeiT-III

Although the addition of registers has a limited effect on the accuracy of classification, it mitigates the existence of artifacts to some extent, improving the interpretability of the model. Figure 7 illustrates that registers can cause artifacts to disappear, although this is not always the case.

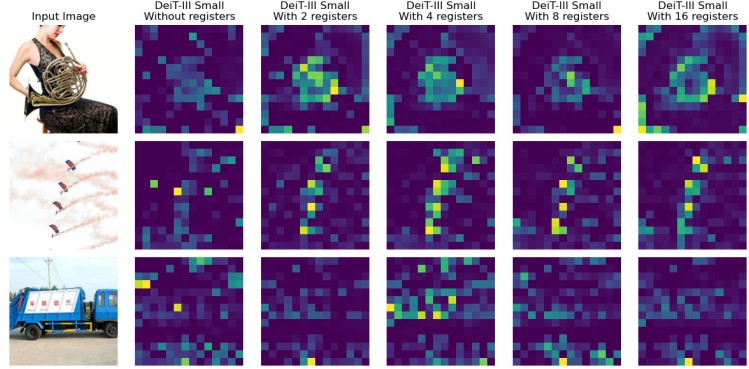

Figure 7: Attention maps of our retrained DeiT-III model using 0, 2, 4, 8 and 16 registers

Unless otherwise specified, all DeiT-III models discussed in this section employ four register tokens, consistent with the configuration used for DINOv2 by Darcet et al.. In the histograms of token norm distributions, visualized in figures 4c and 4d, the outliers in the non-register version (4c) are less present than in DINOv2 giant (4a), yet they still manifest as a high-norm tail. Adding registers in DeiT-III compresses the distribution, indicating a partial absorption of the global information that these outlier tokens had originally carried.

As shown in Figure 8a, high-norm tokens also appear in DeiT-III, especially in deeper layers where certain tokens accumulate excessive information, mirroring DINOv2 giant. Introducing registers reduces both the frequency and intensity of these outliers (Figure 8b), resulting in a more controlled distribution and fewer

norm spikes. However, high-norm artifacts still occur, suggesting that while registers help balance attention allocation, their effectiveness may vary in smaller models like DeiT-III Small compared to larger architectures.

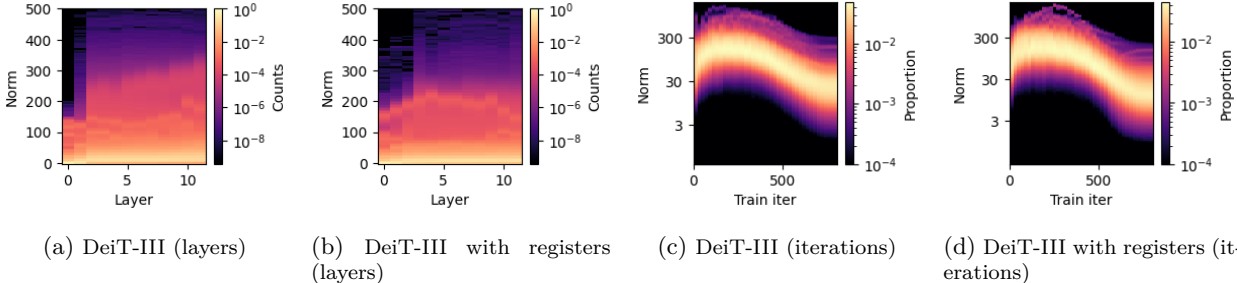

(a) DeiT-III (layers)    (b) DeiT-III with registers (layers)    (c) DeiT-III (iterations)    (d) DeiT-III with registers (iterations)

Figure 8: Distribution of norm values for DeiT-III ((a) and (c)) and DeiT-III with registers ((b) and (d)). The left two plots show norms across layers and the right two plots show norms across training iterations.

To explore how these artifacts relate to the broader token space, we replicate the cosine similarity analysis on DeiT-III (Figure 6b), flagging "artifact" tokens as the top 2% highest norms. Like in the larger model, these high-norm patches appear in uniform, low-texture regions. While outliers capture global cues, they show weaker local alignment with neighboring patches.

This effect is further illustrated in the attention maps of Figure 1. Unlike in DINOv2 giant, where registers consistently improved interpretability, the effect in DeiT-III is more mixed. Some attention maps become more structured with registers, while others retain scattered and concentrated activations. This variability indicates that while registers can mitigate artifacts, their effectiveness may vary depending on specific input images and training dynamics.

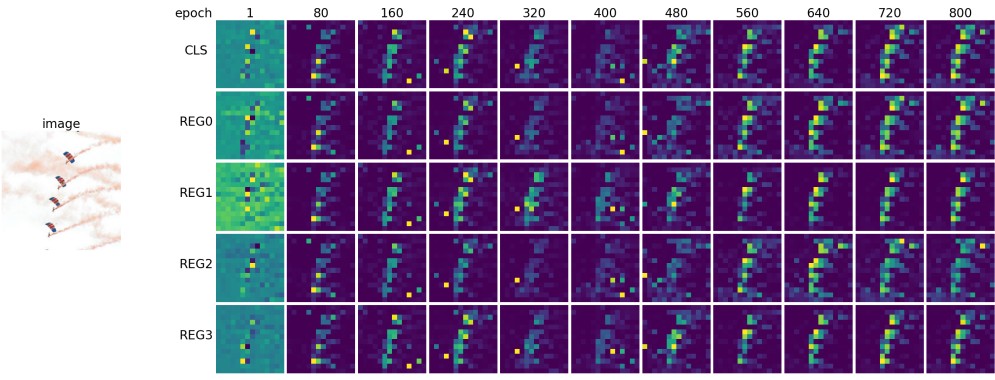

Figure 9: Attention maps of the CLS and register tokens across training epochs where artifacts are successfully removed.

In figure 8, we further analyze the norm distribution of the last layer throughout the training process. We compare models trained without registers (Figure 8c) and with registers (Figure 8d). Although the distribution of norm values appears visually similar for both models, DeiT-III with registers shows visually more compact distributions in the last quarter of training iterations. We did not compute statistical dispersion metrics for this observation, so it should be interpreted as a qualitative trend. However, the model still shows significant outliers in norm values in the first half, indicating that the registers are limited in their ability to combat high-norm tokens.

Figure 9 tracks the evolution of attention maps across training epochs. Here, it can be seen that the model alternates between high-norm outliers and well-structured attention maps. This pattern indicates that attention dynamics are not entirely stable throughout training, yet the registers are often able to mitigate persistent artifacts.

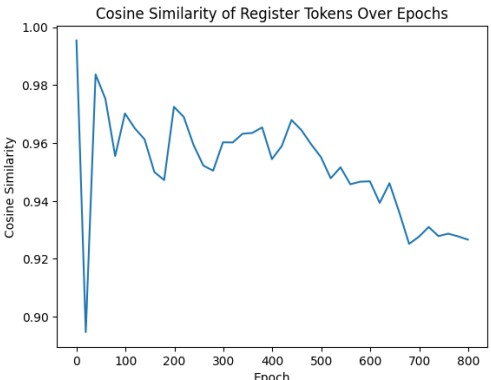

Figure 10: Averaged attention maps for DeiT-III model with registers on the Imagenette dataset. The visualization shows the positional focus of the [CLS] token and register tokens.

Figure 10 shows that in the DeiT-III model with registers, attention remains centered around the [CLS] and register tokens, helping stabilize focus but not completely eliminating local artifacts. We further measured the cosine similarity of attention maps for each register token (Figure 11) and found that, while initially overlapping, these maps diverge slightly over time. However, this divergence is modest, and the attention patterns of the register tokens remain broadly similar. This suggests that, in our DeiT-III Small model, the register tokens do not exhibit clear specialization in their spatial focus.

Figure 11: Mean cosine similarity between all register token attention map pairs over epochs. Lower similarity over time indicates increasing specialization among registers.

Overall, these results highlight that while registers help reduce high-norm artifacts in DeiT-III, their impact is more inconsistent than in DINOv2 giant. Some improvements in attention interpretability are evident, while certain cases still exhibit instability, particularly in deeper layers and across training epochs.

### 4.3 Global Features

### 4.3.1 Object Detection with LOST

In this section, we evaluated ViTs with registers using the LOST (Siméoni et al., 2021) method on the VOC2007 dataset (Everingham et al., 2010). Table 1 shows the CorLoc values for several ViT architectures, including the large and small DINOv2 models and a DeiT-III model with two, four and six registers. Results for DINO (Darcet et al., 2024) are also presented for reference.

For DINOv2, we observe a moderate improvement in the large model, where CorLoc increases from 33.77% to 35.24% with registers. This trend is consistent with the findings in Darcet et al. (2024), though our observed gain is smaller. In contrast, the small DINOv2 variant sees a drop from 37.48% to 32.93%, suggesting that limited capacity may hinder the effective use of register tokens for object localization.

For the DeiT-III configurations, all models share the same baseline without registers (15.13%). As shown in Table 1, the CorLoc score improves progressively with more registers, reaching a maximum of 19.01% with sixteen tokens. These results suggest that adding more registers can enhance localization performance in small ViTs, although the gains are modest and may plateau beyond a certain point.

Table 1: CorLoc values of LOST on different ViT models with and without registers.

| Model | Without Registers (%) | With Registers (%) | Δ (%) |
|---|---|---|---|
| DINOv2 (large) | 33.77 | 35.24 | +1.47 |
| DINOv2 (small) | 37.48 | 32.93 | −4.55 |
| DINO | 61.48 | – | – |
| DeiT-III (2 regs) | 15.13 | 15.49 | +0.36 |
| DeiT-III (4 regs) | 15.13 | 17.92 | +2.79 |
| DeiT-III (8 regs) | 15.13 | 18.03 | +2.90 |
| DeiT-III (16 regs) | 15.13 | 19.01 | +3.88 |

We note that our LOST implementation is fully deterministic: it uses fixed resizing and center cropping, and contains no stochastic components such as dropout or random sampling. As a result, CorLoc scores remain exactly reproducible across runs, even when changing the random seed ($\sigma = 0$).

### 4.3.2 Estimation of global information in spatial patch tokens

Another important aspect of the artifacts described by Darcet et al. (2024) is their potential to encode global information. In their study, the authors theorized that high-norm tokens might carry broad, high-level information crucial for global computations. To investigate this hypothesis, they trained a logistic regressor to classify images based on individual patch tokens. Notably, high-norm tokens consistently outperformed low-norm tokens, indicating that these "artifact" tokens capture more global concepts of the image.

Table 2: Classification accuracy (± std) for our DeiT III model, across three runs.

| | Caltech101 | | CIFAR10 | |
| | no Registers | w/ Registers | no Registers | w/ Registers |
|---|---|---|---|---|
| High Norm | $.8540 \pm .0078$ | $.8153 \pm .0046$ | $.7853 \pm .0067$ | $.7841 \pm .0074$ |
| Low Norm | $.7161 \pm .0062$ | $.7941 \pm .0058$ | $.7534 \pm .0103$ | $.7727 \pm .0030$ |
| [CLS] | $.9988 \pm .0002$ | $.9990 \pm .0004$ | $.9251 \pm .0017$ | $.9285 \pm .0034$ |

To replicate this, we conducted a similar experiment on our DeiT-III model. In each batch, we identified the top 2% of tokens by norms as high-norm outliers and the rest as low-norm tokens. We then selected one high-norm token, one low-norm token, and the CLS token to train a logistic regressor to predict the image class. Table 2 shows that high-norm tokens constantly achieved higher accuracy than low-norm tokens in models without register, but much more similar accuracies in models with registers.[3] This supports the idea that, without registers, artifact tokens encode more global information, and by introducing registers, we can distribute the information more evenly across tokens all.

## 5 Discussion

Our primary aim was to examine whether the main findings of Darcet et al. (2024) hold true for smaller-scale Vision Transformers, specifically a DeiT-III Small model trained on ImageNet-1k. This involved addressing three key questions: (1) Do high-norm artifact tokens also emerge in smaller ViTs? (2) Can registers effectively mitigate these artifacts in a low-parameter regime? and (3) Do the local-global encoding improvements observed in larger ViTs generalize to smaller architectures? Below, we contextualize our observations and connect them back to these questions.

**High-Norm Artifacts in Smaller Models:** Our experiments confirm that, while artifacts do not appear frequently enough in DINOv2 small models, high-norm artifact tokens do arise in DeiT-III Small, albeit less

---

[3]Find additional datasets and results of DINOv2 small in the Appendix C

frequently and less pronounced than in larger models. This finding refines the conclusion from Darcet et al. (2024), whose Figure 4c suggests that only ViT-L and above develop high-norm artifacts. While it is clear that increasing capacity amplifies the effect in DINOv2, our results underscore that smaller architectures are not entirely immune. In other words, although the smaller DeiT-III model exhibits fewer "extreme" outlier tokens, we still observe instances of disproportionately high norms, indicating that artifact formation can emerge even outside the large-model regime.

**Impact of Registers in Mitigating Artifacts:** We found that adding registers generally reduced the prevalence and intensity of these outliers in DeiT-III, echoing the trends observed in DINOv2 giant. While the improvement was not as pronounced as in the original paper, our results do show that registers are beneficial for smaller models as well. Specifically, both norm distributions and qualitative attention maps indicated fewer extreme spikes, aligning with the second key question about the role of registers in stabilizing smaller ViTs.

**Local-Global Encoding and Downstream Performance:** A key question was whether the enhanced local-global encoding capacities reported in Darcet et al. (2024) would extend to smaller models. Using LOST (Siméoni et al., 2021) as a proxy for object-level understanding, we observed moderate CorLoc gains when registers were added to both DINOv2 large and DeiT-III small, while DINOv2 small exhibited a decline in CorLoc performance. Although these gains are more modest than the improvements documented by Darcet et al. (2024) for larger ViT variants, our findings suggest that the effectiveness of register tokens is not determined by model scale alone. Instead, the improvements appear linked to the presence of high-norm artifacts in the attention maps: both DINOv2 Large and DeiT-III Small, which exhibit such artifacts, benefited from registers, whereas DINOv2 Small, which does not, experienced degraded performance.

## 5.1 Limitations

While our reproducibility study confirms that register tokens can mitigate some high-norm artifacts in smaller ViTs, several caveats temper the generality of our conclusions.

First, we restricted ourselves to focusing on a single archtecture-dataset pair (DeiT-III-S on ImageNet-1k) and a fixed set of training hyperparameters. Although some experiments were also performed using DINOv2 small, these experiments were done using the pretrained models which were made available by the original authors. The observed behavior may not generalize to other backbone sizes, patch resolutions, optimization strategies, or data regimes (e.g., domain-shifted or multimodal corpora).

Second, we explored only one register initialization scheme and a narrow range of register counts (2–16). We did not perform a systematic search for optimal configurations or examine interactions with other architectural components such as positional encodings or token pruning.

Third, our evaluation focuses on four axes—classification accuracy, token-norm statistics, attention visualizations, and LOST CorLoc—omitting dense prediction tasks (e.g., segmentation, depth) where the original Darcet et al. (2024) reported the largest gains.

Fourth, the observed gains on small models were modest and sometimes inconsistent across images and metrics. This suggests that the register mechanism may be sensitive to training stochasticity or image complexity.

Taken together, these limitations suggest that our results should be viewed as an initial, rather than comprehensive, validation of the register hypothesis at smaller scales.

## 5.2 Future work

While registers reduce high-norm artifacts in attention maps, their effect is not uniform across all images. A key avenue for future research is understanding why some inputs still exhibit artifact-like behavior while others benefit more substantially. Analyzing the relationship between image complexity, token representation, and attention dynamics could offer deeper insights into when and how registers succeed or fail. Ultimately, this could lead to architectural or training refinements that further enhance the robustness of ViTs.

**Acknowledgments**

We would like to thank Jax for posing in figure 3.

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

## A Model Parameters

Table 3: Hyperparameters used for the DeiT-III Small Model on ImageNet-1k

| Hyperparameter | Value/Description |
|---|---|
| **Dataset** | |
| Dataset | ImageNet-1k (ILSVRC 2012-2017) |
| Input Size | 224 |
| Number of Classes | 1000 |
| **Model Architecture** | |
| Model | `deit_small_patch16_LS` |
| Drop Path Rate | 0.05 |
| **Training Settings** | |
| Batch Size | 256 |
| Epochs | 800 |
| Optimizer | `fusedlamb` |
| Learning Rate | $4 \times 10^{-3}$ |
| Warmup Learning Rate | $1 \times 10^{-6}$ |
| Weight Decay | 0.05 |
| Scheduler | `cosine` |
| Warmup Epochs | 5 |
| Unscale Learning Rate | True |
| **Data Augmentation** | |
| Mixup | 0.8 |
| CutMix | 1.0 |
| Color Jitter | 0.3 |
| Repeated Augmentation | True |
| ThreeAugment | True |
| **Regularization** | |
| Reprob | 0.0 |
| Label Smoothing | 0.0 |
| Drop | 0.0 |
| Drop Path | 0.05 |
| **Loss Function** | |
| BCE Loss | True |
| **Miscellaneous** | |
| Evaluation Crop Ratio | 1.0 |
| Seed | 0 |

In Section 3, we described our primary experimental setup for evaluating registers in Vision Transformers. Here, we provide a detailed overview of the training pipeline and hyperparameters used for our DeiT-III Small experiments on ImageNet-1k, closely following the official DeiT-III repository. Table 3 outlines each component of the setup—including data augmentation, optimization and regularization—ensuring transparency and reproducibility. We highlight key choices, such as employing 4 register tokens (when applicable) and adopting an 800-epoch training schedule. These settings closely replicate the strong baselines established by Touvron et al. (2022) while allowing us to isolate and evaluate the impact of registers under consistent conditions.

## B    Extended Analysis of Persistent Attention Artifacts

In the main text (Figure 9), we illustrated a scenario where the model's attention maps, although prone to transient high-norm outliers, often return to well-structured patterns due to the presence of registers. However, Figure 12 provides a contrasting case. Here, the model remains stuck in outlier states for prolonged periods, leading to persistent artifacts. Despite registers being designed to mitigate such issues, these results emphasize that their effectiveness is not guaranteed under all conditions. Factors such as initialization, optimization strategy or dataset characteristics can undermine the stabilizing role of registers. These observations motivate further exploration into methods for reliably stabilizing attention dynamics in more challenging training scenarios.

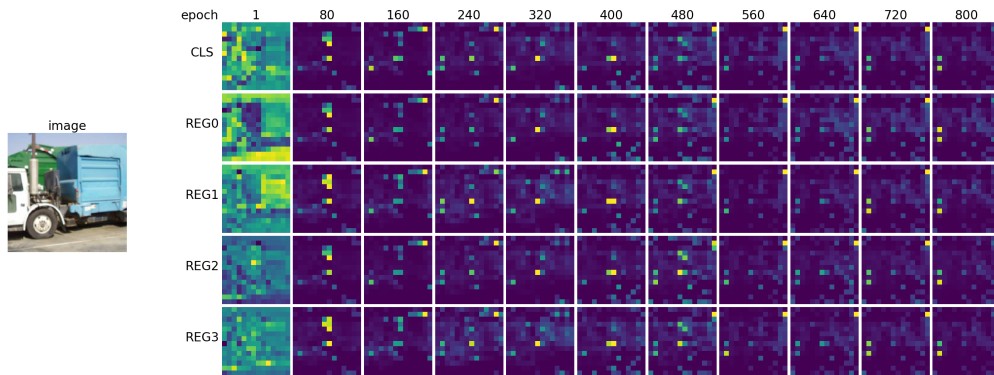

Figure 12: Attention maps of the CLS and register tokens across training epochs in a case where artifacts persist.

## C    Complete Results of Global Classification from Tokens

Table 4: Accuracy results (mean ± std) for High Norm, Low Norm, and CLS tokens of our DeiT III model across more datasets. Configurations were run three times on different initial seeds.

| DeiT III | | | | |
|---|---|---|---|---|
| | **Imagenette** | | **Caltech101** | |
| | no Registers | w/ Registers | no Registers | w/ Registers |
| High Norm | .9568 ± .0076 | .9759 ± .0010 | .8540 ± .0078 | .8153 ± .0046 |
| Low Norm | .9462 ± .0019 | .9781 ± .0036 | .7161 ± .0062 | .7941 ± .0058 |
| [CLS] | 1.0 ± 0.0 | 1.0 ± 0.0 | .9988 ± .0002 | .9990 ± .0004 |
| | **Flower102** | | **CIFAR10** | |
| | no Registers | w/ Registers | no Registers | w/ Registers |
| High Norm | .5608 ± .0319 | .2427 ± .0153 | .7853 ± .0067 | .7841 ± .0074 |
| Low Norm | .1966 ± .0087 | .3180 ± .0217 | .7534 ± .0103 | .7727 ± .0030 |
| [CLS] | 1.0 ± 0.0 | 1.0 ± 0.0 | .9251 ± .0017 | .9285 ± .0034 |

Table 5: Accuracy results (mean ± std) for High Norm, Low Norm, and CLS tokens of the pretrained DINOv2 small model across multiple datasets. Configurations were run three times on different initial seeds.

| | **DINOv2 Small** | | | |
|---|---|---|---|---|
| | **Imagenette** | | **Caltech101** | |
| | no Registers | w/ Registers | no Registers | w/ Registers |
| High Norm | $.9724 \pm .0025$ | $.9859 \pm .0011$ | $.9439 \pm .0048$ | $.9602 \pm .0027$ |
| Low Norm | $.9632 \pm .0015$ | $.9757 \pm .0029$ | $.9123 \pm .0081$ | $.9442 \pm .0075$ |
| [CLS] | $1.0 \pm 0.0$ | $1.0 \pm 0.0$ | $.9994 \pm .0004$ | $.9992 \pm .0008$ |
| | **Flowers102** | | **CIFAR10** | |
| | no Registers | w/ Registers | no Registers | w/ Registers |
| High Norm | $.9000 \pm .0027$ | $.9468 \pm .0180$ | $.9399 \pm .0034$ | $.9485 \pm .0010$ |
| Low Norm | $.7255 \pm .0121$ | $.8720 \pm .0236$ | $.9222 \pm .0038$ | $.9434 \pm .0022$ |
| [CLS] | $1.0 \pm 0.0$ | $1.0 \pm 0.0$ | $.9754 \pm .0026$ | $.9734 \pm .0014$ |

## D   Positional Focus of CLS and Register Tokens on different datasets

Our attention-map visualizations in Section 4 focus primarily on Imagenette. In Figure 13, we extend this analysis to additional datasets—CIFAR10, Caltech101 and Flowers102—to illustrate how registers influence CLS and patch-token attention across diverse visual domains. The CLS token consistently focuses on a central region, reflecting its role in aggregating global information. The register tokens exhibit distinct yet overlapping focus regions. The presence of registers leads to a more distributed attention pattern, suggesting that they share the burden of global information aggregation. While all registers maintain a broad attention spread, subtle variations emerge, indicating potential specialization in capturing different aspects of the image.

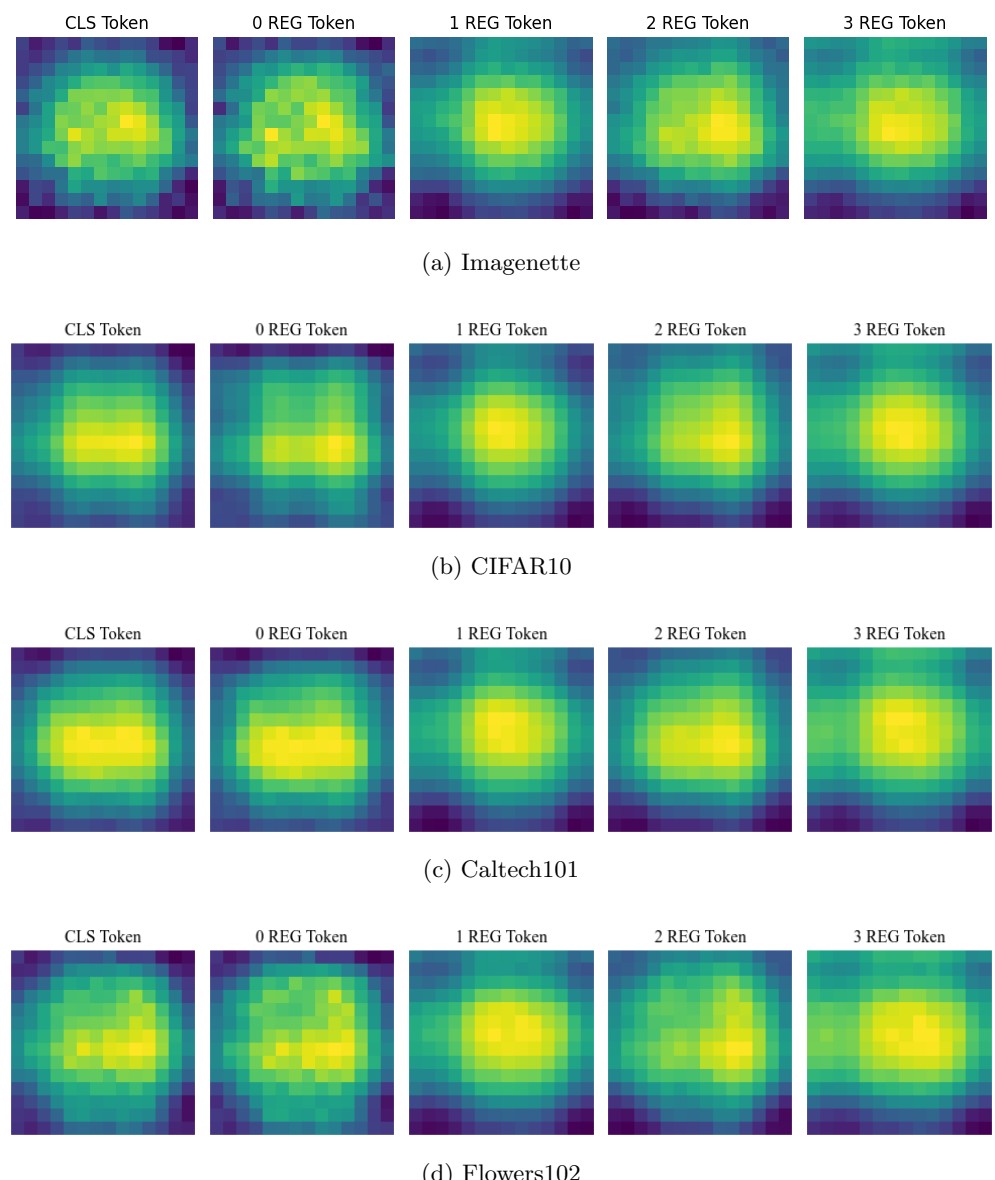

Figure 13: Comparison of positional focus maps for DeiT-III models on different datasets. (a) Imagenette, (b) CIFAR10, (c) Caltech101, (d) Flowers102.

# E    Computational requirements and estimated carbon footprint

For every number of registers tested (0, 2, 4, 8 and 16) the model had to be retrained. Each model was trained on a single node with 4 NVIDIA A100 GPUs and 72 Intel Xeon CPU cores, taking about 66 hours per model (330 hours total). Table 6 summarizes the key parameters used to estimate the carbon footprint. The Power Usage Effectiveness (PUE) is taken from the Snellius documentation[4], while the average Carbon Intensity Factor reflects the Netherlands in 2024. In the table, we list only the approximate per-GPU power draw; CPU usage is not separately itemized. Using these values, we arrive at a total carbon footprint of approximately **256**.4 kg $CO_2$eq for training both models (following Patterson et al. (2021)).

---

[4]https://servicedesk.surf.nl/wiki/display/WIKI/Snellius

Table 6: Summary of carbon footprint for reproducing our model training, following Oquab et al. (2023). Here, *GPU-hours* is the product of the number of GPUs (4) and total hours (330). The power consumption (400 W per GPU) is an approximate average draw, excluding CPU usage.

| GPU Type | GPU Power (W) | GPU-hours | PUE | Total Power (MWh) | Carbon Emitted (tCO$_2$eq) |
|---|---|---|---|---|---|
| NVIDIA A100 | 400 | 1320 | 1.2 | 0.58 | 0.26 |

