# OpenReview forum: "Registers in Small Vision Transformers: A Reproducibility Study of Vision Transformers Need Registers"
_TMLR — Accepted by TMLR_

### Review · Reviewer_JFSv · 2025-04-01

**Summary Of Contributions:**

Inspired by Darcet et al. (2024), this reproducibility study investigates whether smaller ViT architectures also develop high-norm artifacts in their attention maps, and whether adding learnable “register” tokens can mitigate this problem. The authors focus primarily on DeiT-III Small, examining whether:

1. High-norm artifact tokens occur in smaller models
2. Registers can effectively mitigate these artifacts
3. Local-global encoding improvements observed in larger models generalize to smaller architectures

The paper provides qualitative analysis through attention map visualizations and limited quantitative evaluation through classification performance and LOST object discovery benchmarks.

**Audience:**

Yes

**Claims And Evidence:**

Yes

**Requested Changes:**

At the moment, I think the work needs further improvement to meet TMLR’s bar for acceptance.

1. Paper motivation
    - Clarify why studying high-norm attention artifacts in smaller Transformers is important. Should we expect smaller ViTs to be more or less prone to this issue?
    - Discuss your results in relation to the findings from Fig 4c. from [1], which suggest that for DINOv2, only ViT-L and above develop high-norm artifacts.
2. Section 4.2.2: Please clarify what data the token norm is computed on.
3. Experiments
    - Add self-supervised VITs for a different perspective. E.g. is it possible to also train smaller DINO or MAE models? This paper focuses on only DEIT-III small, which is a relatively limited class of models.
    - Quantitative results for more image classification datasets (e.g. IN1K) would provide a better picture of model performance. It is still unclear if adding registers improves results overall.
    - In general, there are no insights about how to best add registers for smaller vision transformers. E.g. what is the optimal amount of registers and how does this change with model size? Is there a saturation point beyond 4 registers?
4. Computational analysis is missing, even if there are only 4 additional tokens.

[1] Vision Transformers Need Registers. Darcet et al. ICLR 2024.

**Strengths And Weaknesses:**

**Strengths:**
- Well structured and written
- Comprehensive visualization of attention maps and token norm distributions
- Confirmation that high-norm artifacts do exist in smaller ViTs, though to a varying degree

**Weaknesses:**

All weaknesses are summarized at a higher level. Specific suggestions are given in the below “Requested Changes” section.

- The motivation for studying registers specifically in smaller Transformers is unclear. The paper doesn’t mention whether we should expect different behavior in smaller models.
- Quantitative results are relatively lacking, and the family of models studied is limited to supervised VITs.
- Limited model family limits scope of insights derived from the work.
- Limited insights and quantitative results.

---

> ### Author Response · Authors · 2025-05-07
> **Implemented Changes requested by Reviewer**
>
> We thank reviewer JFSv for their thoughtful and detailed comments. We appreciate the time taken to evaluate our work and acknowledge the constructive suggestions, many of which we have incorporated into the revised version of our paper. Below, we address each point raised:
>
> ### **Motivation**
>
> > *"The motivation for studying registers specifically in smaller Transformers is unclear..."*
>
> We have revised the Introduction to clarify the motivation: while Darcet et al. (2024) \[1] emphasized that high-norm artifacts predominantly appear in large ViTs (e.g., DINOv2 Large and Giant), smaller models like DeiT-III Small are widely deployed in resource-constrained settings, where lightweight enhancements such as registers could offer practical benefits. Additionally, we cite more recent work (e.g., Nakamura et al., 2024 \[2]; Shang et al., 2024 \[3]) that observed high-norm artifacts in small and medium-sized ViTs, reinforcing the relevance of this question beyond large-scale models.
>
> ### **Discussion of Figure 4c from Darcet et al. (2024)**
>
> > *"Discuss your results in relation to Fig 4c. from \[1], which suggest only ViT-L and above develop high-norm artifacts."*
>
> We now address this directly in the Discussion section (Section 5), stating that our findings refine the conclusion from Darcet et al. (2024): while DINOv2 Small does not consistently exhibit artifacts, DeiT-III Small does, albeit to a lesser extent than its large-scale counterparts. This distinction points to architectural differences that may influence artifact formation.
>
> ### **Clarification on Norm Computation**
>
> > *"Section 4.2.2: Please clarify what data the token norm is computed on."*
>
> We have clarified in Section 4.2.2 that token norms and cosine similarity analyses are computed over the full validation set of the Imagenette dataset (3925 images). This detail is now explicitly stated to improve transparency.
>
> ### **Inclusion of Self-Supervised ViTs**
>
> > *"Add self-supervised ViTs... is it possible to also train smaller DINO or MAE models?"*
>
> We thank the reviewer for this suggestion. While we did not train small self-supervised models from scratch due to resource limitations, we did analyze the pretrained DINOv2 Small model, including attention maps, token norm distributions, and LOST performance. Our findings, summarized in Section 4.2.2 and the Discussion, confirm that DINOv2 Small does **not** exhibit high-norm artifacts. This reinforces the idea that artifacts are architecture-dependent. As such, we refrained from adding registers to DINOv2 Small, as there was no artifact behavior to mitigate. We now emphasize this reasoning more clearly in the revised Discussion.
>
> ### **Quantitative Results and Broader Evaluation**
>
> > *"Quantitative results for more image classification datasets..."*
>
> We have added results for four additional datasets (CIFAR10, Flowers102, Caltech101, and Imagenette) in Appendix C. These evaluate how well individual tokens (high-norm, low-norm, and \[CLS]) capture global class information, both with and without registers. These results support our conclusion that registers help redistribute global information more evenly across tokens.
>
> ### **Optimal Number of Registers**
>
> > *"No insights about how to best add registers... saturation point?"*
>
> We now include an expanded analysis in Section 4.3.1 and Table 1, which explores the impact of 2, 4, 8, and 16 registers on LOST CorLoc. The results indicate a modest but consistent improvement in localization performance up to 16 tokens, without a clear saturation point. This is now discussed more explicitly in Section 5.1 (Limitations), where we also acknowledge the need for a more systematic hyperparameter search in future work.
>
> ### **Computational Analysis**
>
> > *"Computational analysis is missing..."*
>
> Thank you for pointing this out. We now include a carbon footprint analysis in Appendix E, estimating the training cost for all register configurations. This includes GPU hours, power usage, and total emissions. While we do not include inference-time latency or parameter count deltas (which are minor), this section adds transparency about the resource requirements for our experiments.
>
> \[1] Darcet, T. et al. *Vision Transformers Need Registers*. ICLR 2024.
>
> \[2] Nakamura, K. et al. (2024). *Improving Image Clustering with Artifacts Attenuation via Inference-Time Attention Engineering*. arXiv preprint arXiv:2410.04801. [https://arxiv.org/abs/2410.04801](https://arxiv.org/abs/2410.04801)
>
> \[3] Shang, J. et al. (2024). *Theia: Distilling Diverse Vision Foundation Models for Robot Learning*. arXiv preprint arXiv:2407.20179. [https://arxiv.org/abs/2407.20179](https://arxiv.org/abs/2407.20179)

---

### Review · Reviewer_BWaK · 2025-04-19

**Summary Of Contributions:**

This paper seeks to reproduce the paper “Vision Transformers Need Registers” (Darcet et al 2024) as well as instead the investigation of this paper to smaller ViTs. In particular, this paper seeks to reproduce the following phenomena from the original paper:  (1) the existence of artifacts (high-norm tokens) in larger models, (2) that adding registers (additional learnable tokens) mitigates these artifacts, and (3) that ViTs with registers have better intermediate features.

This paper runs experiments using pre-trained DINOv2 models (with 300M parameters) as well as DeiT-III models (with 22M parameters). The pre-trained DINOv2 models included versions without registers and with registers. The DeiT-III models were trained from scratch on ImageNet-1K with both vanilla and register versions. The register versions had four register tokens.

DeiT-III models were examined across several experiments: (1) classification accuracy, comparing models without register to models without registers, where register models had slightly lower but comparable accuracy, (2) local reconstruction using low- and high- norm tokens, (3) object detection with LOST, and (4) image classification on high-norm tokens. They were also analyzed on the distribution of norms with and without registers (Figure 4), distribution of norm layers (Figure 7ab) and across iterations (Figure 7cd), and cosine similarity to neighboring patches for top 2% tokens (Figure 6).

DINOv2 models were examined only on object detection with LOST, and were also analyzed in the same ways as the DeiT-III models, with the exception of the distribution of norms across training iterations.

**Audience:**

Yes

**Broader Impact Concerns:**

I don't believe this paper needs a broader impact statement

**Claims And Evidence:**

Yes

**Requested Changes:**

Main changes:
1. Define high- and low- norm tokens early in the paper
2. Moderate the claim that using registers distributes the accumulated global information of high-norm tokens or provide additional evidence for this claim (such as the experiment suggested)
3. Quantify the claim that Deit-III with registers shows a more compact distribution in the last quarter of training iterations by providing statistics on the distribution
4. Remove the claim that register tokens in DeiT-III with registers have specialized or provide additional evidence for it
5. Quantify the claim that the effect of registers is image-dependent with the percentage of images where artifacts are removed for DINOv2/DeiT-III
6. Provide errors bars (over multiple runs) for Tables 1, 2

**Strengths And Weaknesses:**

Strengths:

This paper examines the phenomenon of artifact tokens in vision transformers and if registers can alleviate them, as described in Darcet et al 2024, for smaller ViTs, and also reproduces some of the results of this paper. The strengths of this paper include the partial reproduction of Darcet et al 2024, tackling the interesting and important problem of whether registers help for small ViTs, and that the paper is clear and well-written.

Weaknesses:

Figure 1 could benefit from being computed in higher resolution (as in the original paper Darcet et al 2024) and perhaps with a color scale.

LOST is introduced on page 4 but not explained and no paper is cited.

In general, it would be better if “high-norm” and “low-norm” was defined early in the paper to allow a better evaluation of claims about high- and low- norm tokens. This would be helpful for understanding the “local information of high-norm tokens” experiment (Section 4.2.1).

I also think the claim about global information accumulation in non-register models and global context redistribution should be moderated, as I don’t think you can conclude this on the basis of Figure 4 alone. I think this claim could be supported quantitatively by repeating the experiment in 4.3.2 with the DeiT-III-register model and showing that classification accuracy for the low-norm tokens increases and the classification accuracy for high-norm tokens decreases.

This claim should be quantified: “Although the distribution of norm values appear visually similar for both models, DeiT-III with registers shows more compact distributions in the last quarter of training iterations”, especially as it’s hard to tell visually that this is the case.

The suggestion that register tokens in DeiT-III have specialized is not very well-supported, relying only on subtle differences in the attention map.

The claim that the effect of registers is image-dependent should be quantified.

For Table 2, since the values can be very close, especially on the Imagenette dataset, I think the claims would be more strongly supported with the inclusion of error bars. The conclusion that “high-norm tokens consistently outperformed low-norm tokens” seems too strong for the evidence presented, which indicates that high- and low-norm tokens perform comparably on Imagenette. It would also be nice to see error bars for Table 1 as well.

---

> ### Author Response · Authors · 2025-05-07
>
> We sincerely thank reviewer BWaK for their thoughtful and detailed feedback. We greatly appreciate the time and effort taken to assess our submission and offer constructive suggestions for improvement. Below, we provide a point-by-point response to the comments raised and outline the corresponding changes made to the revised version.
>
> ### Definition of High- and Low-Norm Tokens
>
> > *"It would be better if ‘high-norm’ and ‘low-norm’ was defined early in the paper..."*
>
> We agree, and thank the reviewer for this suggestion. In the revised version, we now define **high-norm**, **low-norm**, and **register** tokens at the end of Section 1 (under “Key Concepts and Terminology”). We also clarify their relevance throughout the Results sections to better support our claims.
>
> ### Visualization Resolution and Color Scale
>
> > *"Figure 1 could benefit from being computed in higher resolution (as in the original paper...)"*
>
> Due to the fixed input resolution of **224×224** in our training and evaluation pipeline, higher-resolution visualizations were not possible without architectural changes that would have compromised comparability with the original DeiT-III setup.
>
> ### Clarification of LOST and Citation
>
> > *"LOST is introduced but not explained and no paper is cited."*
>
> We have updated Section 3 to include a concise explanation of LOST (Localizing Objects with Self-supervised Transformers) and added the appropriate citation to Siméoni et al. (2021) [1].
>
> ### Moderation and Quantification of Claims
>
> > *"The claim about global information accumulation... should be moderated or supported with quantitative evidence."*
>
> Thank you for pointing this out. We now moderate the language used in Section 4.2.2 and 4.3.2. Specifically, we clarify that the observed redistribution of global information is **suggested** by our findings rather than definitively proven, and we refer to the classification performance of high- and low-norm tokens (Table 2 and Appendix C) to illustrate this trend.
>
> > *"Quantify the claim that DeiT-III with registers shows a more compact distribution..."*
>
> We agree this observation was too vague in the original version. We have updated the discussion in Section 4.2.3 to clearly state that the visual compactness is a **qualitative** trend and added a disclaimer noting that no dispersion statistics were computed.
>
> ### Specialization of Register Tokens
>
> > *"The suggestion that register tokens have specialized is not very well-supported."*
>
> We appreciate the reviewer’s caution and have removed this claim. Section 4.2.3 now focuses solely on observed attention patterns without making inferences about functional specialization, which was indeed not strongly supported.
>
> ### Error Bars and Statistical Support
>
> > *"Provide error bars for Table 2 and Table 1..."*
>
> We have added standard deviation across three seeds for **Table 2**, as requested. For **Table 1**, we now clarify that our LOST implementation is fully deterministic—using fixed center cropping and no random components—resulting in zero variance across runs. We explicitly state this in the caption and body text.
>
> \[1] Localizing Objects with Self-Supervised Transformers and no Labels. Siméoni et al. ICCV 2021.

---

### Review · Reviewer_dYQ4 · 2025-04-23

**Summary Of Contributions:**

The goal of this paper is to investigate the Vision Transformers (ViTs) in a small-scale setting in terms of three following questions: 1) whether high-norm artifact tokens emerge in small ViTs?; 2) whether registers effectively address these artifacts in a low-parameter regime?; 3) whether the local-global encoding improvements in large ViTs can be generalized to small ViTs? To answer these questions, the authors perform several experiments with clear description and reports.

**Audience:**

Yes

**Claims And Evidence:**

Yes

**Requested Changes:**

1. The authors should include a paragraph in Section 5 to acknowledge the limitations of their work.

**Strengths And Weaknesses:**

**Strengths:**

1. The investigation into the registers in Vision Transformers (ViTs) is of interest and lays the ground for improving the ViTs performance.

2. The experiments are conducted thoroughly with sufficient details.

3. The paper is well-written and easy to follow. The problem of interest is clearly stated.

**Weaknessess:**

1. While I agree that the investigation into the registers in Vision Transformers (ViTs) is of interest, I am not sure if the study of registers in small-scale ViTs (rather than large-scale ViTs) is of high importance.

2. The authors should include a preliminary section to introduce the definition of some specific terms like high-norm artifacts, registers as well as provide corresponding instances so that readers from other background can understand.

---

> ### Author Response · Authors · 2025-05-07
>
> We thank reviewer dYQ4 for taking the time to read and evaluate our paper, and for their positive remarks regarding the clarity, structure, and thoroughness of our experiments.
>
> > **"I am not sure if the study of registers in small-scale ViTs... is of high importance."**
>
> We agree that this question is not settled a priori. However, small ViTs are widely used in edge devices and resource-constrained environments. If high-norm artifacts (as identified by Darcet et al., 2024) also emerge in these models, then registers could offer a lightweight architectural remedy. As discussed in Section 1 (Introduction) and revisited in Section 5 (Discussion), our findings suggest that artifact formation is not limited to large models—DeiT-III Small does exhibit high-norm tokens—and that registers can partially mitigate these. Thus, exploring the utility of registers in this regime contributes toward a more complete understanding of their applicability.
>
> > **"The authors should include a preliminary section to introduce the definition of some specific terms..."**
>
> Thank you for this helpful suggestion. We now include Section 1.1 **Key Concepts and Terminology**, where we define *high-norm tokens*, *low-norm tokens*, and *registers*, as well as briefly explain their role in attention maps. We also provide visual illustrations in Figure 1 to help contextualize these concepts for readers from diverse backgrounds.
>
> > **"The authors should include a paragraph in Section 5 to acknowledge the limitations of their work."**
>
> We have added a full subsection, **Section 5.1 Limitations**, where we discuss the scope and boundaries of our findings. These include the use of a single architecture–dataset pair, the absence of dense prediction tasks, and the modest and sometimes inconsistent gains in smaller models. We hope this addition clarifies the generalizability of our conclusions and aligns with the reviewer’s expectations.

---

### Review · Reviewer_RC8a · 2025-04-28

**Summary Of Contributions:**

This paper presents a reproducibility study of "Vision Transformers Need Registers." The study aims to extend the findings regarding high-norm artifact tokens in Vision Transformers and the use of register to small transformers. The authors conducted experiments on the DeiT-III Small model trained on ImageNet-1k to evaluate whether the benefits of registers observed in larger models persist in a smaller context. The authors report that the benefits of registers in small ViTs are smaller than in larger ViTs.

**Audience:**

Yes

**Broader Impact Concerns:**

It would be beneficial for the authors to include a Broader Impact Statement discussing the potential implications of their work, such as the application of improved small ViTs in resource-constrained or sensitive area.

**Claims And Evidence:**

Yes

**Requested Changes:**

1. Include additional small ViT architectures in the study, such as CCT [1], EfficientFormer [2] and other small ViTs, to assess whether the benefits of registers generalize across different model designs.
2. Perform multiple runs with different random seeds for each experiment and report the results with statistical measures (e.g., mean ± standard deviation).
3. Given the different impacts of registers, provide a more in-depth comparison between DeiT-III Small and DINOv2 by analyzing architectural components such as depth, width, and embedding dimensions.
4. Conduct ablation studies on the number of register tokens to investigate the sensitivity of the results.

**Strengths And Weaknesses:**

**Strengths**
1. The paper addresses an important aspect of model scalability, evaluating whether the advantages of registers in large ViTs apply to smaller models.
2.  The authors provide access to their codebase, facilitating reproducibility and further research.
3. The paper is well-organized and easy to follow.

**Weakness**
1. The study’s conclusions are based solely on the DeiT-III Small model. To generalize the findings, it would be beneficial to include a broader range of small ViT architectures, such as Compact Convolutional Transformers [1] and EfficientFormers [2], and other specifically designed Transformers for efficiency and performance in resource-constrained environments.

2. The experimental results are reported without measures of variability (e.g., mean ± standard deviation). It would be better to conduct experiments multiple times with different random seeds and report the mean and standard deviation to evaluate the robustness of the findings.

3. According to the paper, the benefit of registers in DeiT-III Small is much smaller compared to DINOv2. However, the paper does not discuss the reasons behind this difference in detail. Rather than attributing it simply to model size differences, it would be beneficial to conduct ablation studies on depth, dimension, and other architectural aspects.

4. Some additional ablation studies would strengthen the work. For example, the study fixes the number of register tokens without exploring the sensitivity of results to this choice.

Reference:
[1] Hassani, Ali, et al. "Escaping the big data paradigm with compact transformers." arXiv, 2021.
[2] Li, Yanyu, et al. "EfficientFormer: Vision transformers at MobileNet speed." NeurIPS, 2022.

---

> ### Author Response · Authors · 2025-05-07
>
> We thank reviewer RC8a for their thoughtful comments and for highlighting both the strengths and areas for improvement in our work. Below we address the main points raised.
>
> **Scope of Architectures**
>
> > *"Include additional small ViT architectures in the study, such as CCT \[1], EfficientFormer \[2]..."*
>
> We appreciate this suggestion. Our study was scoped around DeiT-III Small to allow a controlled and focused investigation into the artifact behavior observed in larger ViTs by Darcet et al. (2024) \[3]. DeiT-III offers a strong supervised baseline and an architecture structurally similar to larger ViTs, making it a suitable test for evaluating register effectiveness in a low-parameter regime. Exploring the role of registers in other compact transformer architectures like CCT \[1] and EfficientFormer \[2] is a promising direction for future work but was beyond the current study's scope.
>
> **Statistical Robustness**
>
> > *"Perform multiple runs with different random seeds..."*
>
> Thank you for this important suggestion. We now include mean and standard deviation across three seeds for all experiments where stochasticity is involved (see Tables 2, 4 and 5). As noted in Section 4.3.1, our implementation of LOST is deterministic, so we did not report variance for those results.
>
> **Architectural Comparisons Between DINOv2 and DeiT-III**
>
> > *"...provide a more in-depth comparison between DeiT-III Small and DINOv2..."*
>
> We agree that understanding why register tokens are more beneficial in some architectures than others is a valuable question. In the revised Discussion, we elaborate on how architectural differences might explain why high-norm artifacts were absent in DINOv2 Small but present in DeiT-III Small. Since artifacts must be present for registers to mitigate them, this explains why we refrained from modifying DINOv2 Small further.
>
> **Sensitivity to Register Count**
>
> > *"Conduct ablation studies on the number of register tokens..."*
>
> We did explore varying the number of registers (2, 4, 8, and 16) in DeiT-III and report their impact on LOST CorLoc in Table 1. We observed modest improvements with more registers. We have made this clearer in Section 4.1. A deeper study of register interaction with other hyperparameters (e.g., training schedule, pruning) is left to future work.
>
> **Broader Impact**
>
> > *"Include a Broader Impact Statement..."*
>
> Thank you for this suggestion. In Section 2.3, we discuss potential applications of our findings in low-resource settings and the relevance of interpretability tools for understanding the behavior of small ViTs deployed in sensitive or constrained environments.
>
>
> \[1] Hassani et al., “Escaping the big data paradigm with compact transformers,” *arXiv*, 2021.
>
> \[2] Li et al., “EfficientFormer: Vision transformers at MobileNet speed,” *NeurIPS*, 2022.
>
> \[3] Darcet et al., “Vision Transformers Need Registers,” *ICLR*, 2024.

---

> > ### Comment · Reviewer_RC8a · 2025-05-24
> >
> > Thank you for the authors' response. They have addressed some of my concerns, however I still believe it is important to extend the experiments to other Transformer architectures—at least to some commonly used small Vision Transformers (ViTs) beyond DeiT. This broader evaluation is essential to support the conclusions drawn in Registers for Small Vision Transformers, as emphasized in the title.

---

### Decision · Action_Editor_jxjc · 2025-06-10

**Recommendation:** Accept with minor revision

**Audience:**

Yes

**Audience Explanation:**

The paper has done a solid contribution to the ongoing discussion on architectural enhancements for Vision Transformers by exploring the utility of register tokens in small ViTs. This is particularly relevant for researchers and practitioners working with resource-constrained models or seeking to improve interpretability and performance without substantially increasing model complexity. The study is focused on reproducibility and is likely to be of interest to those investigating efficient model design and Transformers' inner-workings.

**Claims And Evidence:**

Yes

**Claims Explanation:**

The submission investigates whether high-norm attention artifacts observed in large Vision Transformers (ViTs) also occur in smaller models, and whether auxiliary register tokens mitigate such artifacts. The authors present a clear experimental methodology, replicate findings from the paper, and supplement their conclusions with both qualitative (e.g., attention visualizations) and quantitative (e.g., LOST for localization, classification accuracy) results. Overall, this is an empirically driven study and the claims are well-supported by experimental evidences.